# Retrospective Analysis of Real-World Data for the Treatment of Obstructive Sleep Apnea with Slow Maxillary Expansion Using a Unique Expansion Dental Appliance (DNA)

**Nhi Dao** [1]**, Colette Cozean** [2]**, Oleg Chernyshev** [3]**, Clete Kushida** [4]**, Jonathan Greenburg** [5] **and Jonathan S. Alexander** [1,]*

1 Department of Molecular and Cellular Physiology, Louisiana State University Health Shreveport, Shreveport, LA 71103, USA; hda001@lsuhs.edu
2 EyeDeas Company, Lake Forest, CA 92630, USA; colettecozean@gmail.com
3 Department of Neurology, Louisiana State University Health Shreveport, Shreveport, LA 71103, USA; oleg.chernyshev@lsuhs.edu
4 Stanford Sleep Medicine Center, Redwood City, CA 94063, USA; clete@stanford.edu
5 SnoreExperts, Toluca Lake, CA 91602, USA; jgreenburg@outlook.com
* Correspondence: jalexa@lsuhsc.edu; Tel.: +1-318-675-4151

**Abstract:** In addition to mandibular advancement devices, dental expansion appliances are an important clinical approach for achieving an increased intra-oral space that promotes airflow and lessens the frequency or severity of apneic events in patients diagnosed with obstructive sleep apnea (OSA). It has been thought that dental expansion in adults must be preceded by oral surgery; however, in this paper, we examine the results of a new technique for slow maxillary expansion without any surgical procedures. The palatal expansion device, DNA (Daytime-Nighttime Appliance), was reviewed in this retrospective study, particularly regarding its effects on measurements of transpalatal width, airway volume, and apnea-hypopnea indices (AHI) as well as its common modalities and complications. The DNA effectively reduced AHI by 46% ($p = 0.00001$) and significantly increased both airway volume and transpalatal width ($p < 0.00001$). After DNA treatment, 80% of patients showed some improvement in AHI scores with 28% of patients having their OSA symptoms completely resolved. Compared to the use of mandibular appliances, this approach is intended to create a sustained improvement in airway management that can reduce or eliminate dependence on continuous positive airway pressure (CPAP) or other OSA treatment devices.

**Keywords:** obstructive sleep apnea; mandibular advancement device; maxillary expansion device



## 1. Introduction

Obstructive sleep apnea (OSA) is a highly common sleep disorder that affects at least 22 million individuals in the United States. OSA is usually caused by recurrent, transient, anatomic obstructions of the upper airway during sleep, which provoke intermittent hypoxemia (defined as an $sPO_2 \leq 90$). Hypoxemia in OSA produces deleterious effects on sleep quality by provoking arousals, sleep fragmentation, and intermittent sympathetic nervous system activation [1]. Intermittent hypoxemia results from increased sympathetic activity and is additionally associated with reductions in cerebral blood flow, dysregulated cerebral autoregulation, impaired endothelial function, increased intracerebral pressure, and platelet activation, inflammation, and oxidative stress, all of which exacerbate forms of neurological and cardiovascular injuries [2]. With these contributing factors, OSA has been convincingly linked to the intensification of several comorbid conditions, including cardiovascular disease [3–5], earlier mortality [6,7], cancer [8], insulin resistance [9], obesity [10], and greater stroke incidence [11,12]. Approximately 80% of OSA cases are considered 'moderate to severe,' and the disorder is highly underdiagnosed. OSA affects

30% of the male and 15% of the female population with an estimated economic cost that exceeds $150 billion per year within the United States [13,14]. Far from being benign, OSA and its constellation of co-morbidities represent a serious and often unrecognized, but largely treatable, clinical entity.

Compared to continuous positive airway pressure (CPAP) therapy, the management of OSA using most oral appliances is not as effective for treatment [15,16]. CPAP performs better than either mandibular advancement devices (MAD) or tongue-retaining devices [17] for the management of OSA, and, in adults, CPAP therapy remains the most common OSA treatment, reflecting its 95% 'cure' rate. Oral appliances remain the most commonly used alternative to positive airway pressure approaches, and patient adherence to oral appliances exceeds that of CPAP, reflecting the mechanical complexity, discomfort, and high maintenance associated with CPAP. In comparison, mandibular advancement devices are smaller, less cumbersome, and less obtrusive while still maintaining a patient compliance rate of ~80% [18]. As such, MADs remain the leading alternative to CPAP in patients seeking airway management for OSA.

In children, the surgical resection of tonsils and adenoids is often an initial and effective option for OSA [19]. Although CPAP therapy can treat OSA in children as effectively as in adults, CPAP compliance in children and adolescents is remarkably low (51% non-adherent to therapy) [20]. In comparison, orthodontic approaches, e.g., rapid maxillary expansion (RME), may represent more successful options, particularly when used in conjunction with tonsil/adenoid removal to improve airway management. RME devices treat OSA by expanding the oral airway and remodeling the maxilla to increase airflow. These appliances are customized according to the patient's oral anatomy and achieve expansion by applying centrifugal forces against the patient's upper molars. When observing the effects of maxillary expansion for other indications, several dentists reported that their patients also noticed significant improvements in snoring, sleep apnea, and other symptoms of sleep-disordered breathing. It is hypothesized that treatment increases the pharyngeal space, allowing for better airflow and reduced snoring. Vinha et al., found that surgically assisted rapid maxillary expansion in adults significantly reduced the respiratory disturbance index by an average of 54.6% ($p = 0.0013$). The apnea-hypopnea index (AHI) in that study was significantly reduced by 56.2% ($33.23 \pm 39.5$ to $14.5 \pm 19.4$, $p = 0.001$), and Epworth sleepiness scale (ESS) scores also increased from $12.5 \pm 5.3$ to $7.2 \pm 3.5$ ($p < 0.001$) [21]. In another study of mandibular and maxillary expansion in children, Remy et al. found that OSA symptoms significantly improved after nine months, with the AHI significantly decreased by 53% overall and 63% in the youngest participants [22]. Because adult oral anatomy is assumed to be 'fixed,' most expansion appliance studies in adults use surgically assisted RME, which, although more rapid, has the potential to cause pain that requires management and surgery [23,24].

This retrospective study evaluates the efficacy of the DNA (Daytime-Nighttime Appliance) slow maxillary expansion device without surgery as an OSA treatment that produces persistent improvements in AHI, transpalatal width, and airway volume after treatment, as measured in sleep studies of DNA-treated individuals not wearing the device.

## 2. Materials and Methods

### 2.1. Study Purpose

The origin of this study was a request by the Food and Drug Administration for additional real-world data using a DNA appliance with maxillary expansion as the mechanism of action to treat OSA. In examining data from a five-year period between January 2018 and April 2022, this retrospective analysis utilized a research database that was initiated in 2018. Within the database, doctors were allowed to update information throughout the patient treatment process. After IRB exemption by the Program for Protection of Human Subjects at the Icahn School of Medicine at Mount Sinai (STUDY-21-01561), we conducted a retrospective review of the Vivos Airway Intelligence Service (AIS) database, a prospectively collected clinical database. Patient data were anonymized for use by the research group.

Data entry into the AIS was performed by participating dental providers. Demographics, pre- and post-treatment sleep study parameters, and cone beam CT (CBCT) parameters for trans-palatal width (TPW) and total 3-dimensional (3D) airway volume were recorded. This study evaluated real-world data to examine the extent to which the DNA device reduced symptoms of obstructive sleep apnea as measured using decreases in AHI and increases in pharyngeal airway and transpalatal width. Measurements were taken pre- and post-treatment, with each patient serving as their own control when not wearing the DNA device.

### 2.2. Inclusion/Exclusion Criteria

The database was filtered using the following criteria:

1. Participants were adults (18 years or older).
2. Participants had a diagnosis of OSA (AHI > 5).
3. Participants were treated with the DNA Appliance (distributed by Vivos Therapeutics, Inc., Littleton, CO, USA).
4. Pre- and post-treatment AHI scores were at least six months apart.
5. Participants were required to wear the appliance for at least 10 h per day.

The Vivos method includes several adjunctive therapies used at the dentist's discretion in addition to appliance therapy, such as myofunctional therapy; surgical releases; and CPAP during the onset of treatment if the patient presents with moderate to severe clinical symptoms. Seven patients were excluded from the study as a result of concurrent adjunctive therapies, for which concurrent oral appliances are contraindicated and can potentially confound the interpretation of these outcomes (3 for laser surgery, 1 for tonsillectomy, and 3 for transitioning to MAD devices, of which 1 patient also had sinus surgery). In total, 87 patients were considered in the final analysis (Tables 1–3). AHI results are also presented with no adjunct therapies and in combination with adjunct therapies.

**Table 1.** Patient Demographics.

| Parameter | Average | Range |
|---|---|---|
| Age in Years | $48.7 \pm 14.6$ years | 19–84 years |
| BMI Pre-treatment—($kg/m^2$) | $26.7 \pm 5.5$ $kg/m^2$ | 15.2–39.1 $kg/m^2$ |
| BMI Post-treatment—($kg/m^2$) | $26.1 \pm 5.4$ $kg/m^2$ | 15.9–39.0 $kg/m^2$ |
| Length of Treatment | $14.8 \pm 6.7$ months | 5–36 months |
| Average Daily Wear Time | $12.4 \pm 2.7$ h | 10–20 h |

**Table 2.** Gender Breakdown.

| Male | Female |
|---|---|
| 39 (45%) | 48 (55%) |

**Table 3.** Ethnicity.

| Ethnicity | Count | Percent |
|---|---|---|
| Caucasian | 74 | 85% |
| Asian | 2 | 2% |
| Hispanic or Latino | 3 | 4% |
| Black or African American | 1 | 1% |
| Native Hawaiian | 1 | 1% |
| Other | 6 | 7% |

### 2.3. Device Design

The DNA appliance comprises two upper and lower customized dental trays. With the addition of standard orthodontic acrylics and wires for retention, the device can be

adjusted antero-posteriorly (AP), transversely (TV), or within the vertical dimension of occlusion (VDO) (Figure 1). An optional extender on the device further aided in opening the pharyngeal airway. The amount of acrylic utilized for each appliance depended on a consultation from the prescribing dentist for complete coverage of the patient's palate. The appliance was used daily in the evenings or at nighttime. Additionally, at regular intervals (usually once a week, but at least once a month), the patient or dentist expanded the appliance by adjusting the expansion screw by 0.25 mm, as prescribed. For this study, patients were required to wear the device for at least 10 h per day for any duration between 6 and 24 months. Treatment was concluded when there was stabilization or improvement in AHI and adequate space for teeth to straighten with post-treatment alignment. During treatment, patients also had the option to be treated with adjunctive OSA treatments, such as CPAP and myofunctional therapy.

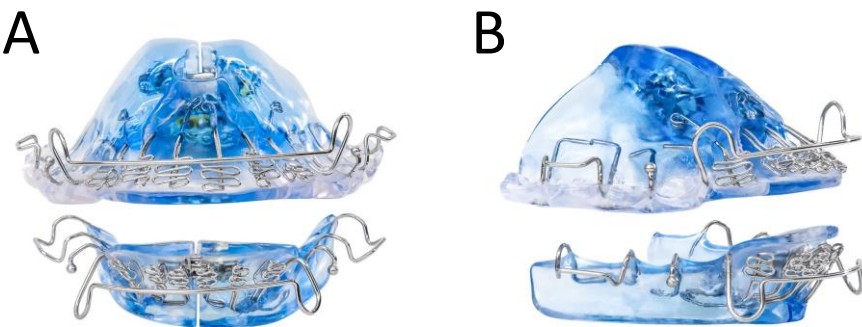

**Figure 1.** (**A**) Direct frontal view of a daytime-nighttime appliance (DNA); (**B**) Left-frontal view of the DNA.

### 2.4. Slow Maxillary Expansion compared to Rapid Maxillary Expansion as a Treatment for OSA

Palatal expansion with an orthodontic device has been shown to increase the pharyngeal space and the size of the airway. Rapid maxillary expansion following surgery to release the tendon is a well-known treatment in orthodontics with adults; however, many patients are reluctant to undergo the required surgery involved with this technique, and studies have shown that the treatment can have severe complications [25]. In this protocol, the patients underwent slow maxillary expansion (1 turn/week on the expansion screw, i.e., 0.25 mm) without any surgery. The expansion of the maxilla was hypothesized to allow more air to pass through the airway and prevent the vibration that causes snoring, or a closure of the airway, in OSA.

### 2.5. Safety and Efficacy Measurements

Sleep studies (polysomnograms (PSGs) or home sleep tests (HSTs) using FDA cleared devices) were conducted prior to and following treatment to assess outcomes without any device in the mouth. Sleep studies were recommended within 1 year before the start of treatment and continually monitored every six months. At each interval, dentists measured the transpalatal width of the jaw at the first upper molar at the closest point between the lingual surfaces and conducted a cone beam computed tomography (CBCT) scan. The CBCT scans were sent to an independent, certified reader. Airway volume was measured at a defined region of the pharynx (excluding nasal passages). A 3D voxel selection was made within a specified boundary around the pharynx with the inferior airway border parallel to both the floor and the anterior-inferior border of C3 and flowing along the anterior nasopharynx. Its length was measured from the top of the pharynx to the inferior border of C3. Subsequently, the dentist filled out a questionnaire about safety concerns for each patient. Both the sleep study measurements and CBCT scores were then reviewed by outside sleep experts, minimizing bias and enhancing reliability.

For this study, almost all AHI measurements were performed using home sleep studies, with less than 5% completed using PSG, comparable to the measurements in clinical care

and research. Additionally, 93% of initial sleep tests were performed less than one year out from the start of treatment, with the rest performed between 18 and 24 months before the start of treatment and only one performed a month following the start of treatment. On average, the sleep study was performed 4 months prior to the start of treatment, including the time to measure, make, and fit the custom device.

All measurements were conducted without the device in the mouth, including the sleep study measurements (PSG and HST). For CBCT measurements, the airway cross-sectional area was measured at the narrowest point of the patient's upper airway (see Figure 2). Additionally, several safety parameters were evaluated, including pain, open bite, excessive salivation, periodontal disease, tissue swelling, among other complications.

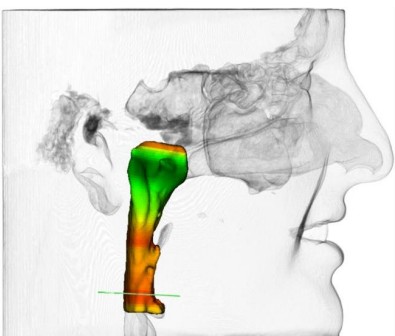

**Figure 2.** Airway volume was measured at a defined region of the pharynx (excluding nasal passages) (green area).

The following endpoints were considered in this retrospective analysis:

1. Pre-treatment to post-treatment improvement in AHI as measured using PSG or HST.
2. Pre-treatment to post-treatment increase in transpalatal width.
3. Pre-treatment to post-treatment increase in airway volume as measured using CBCT.
4. Safety parameters.

### 2.6. Statistical Analysis

Statistical analyses were performed on all data sets using a basic Student's paired, two-sided *t*-test that compared pre- and post-treatment measurements with each patient acting as their own control. With a power of 80%, a confidence level of 95%, an alpha-value (possibility of Type I error) of 0.05, and an assumed improvement of 50%, it was estimated that approximately 40 patients would provide statistically significant results. The sample size was sufficient to achieve statistically significant results for all endpoints considered.

## 3. Results

### 3.1. Expansion Results

The results of DNA on changes in transpalatal widths are shown below in Table 4.

One (1) out of 76 patients with pre- and post-treatment transpalatal width measurements (1%) showed a decrease in transpalatal width, while two patients (3%) showed no change in transpalatal width. Ninety-six percent (96%) of these patients increased their transpalatal width.

**Table 4.** Transpalatal Width changes (in mm).

| Transpalatal Width Pre-Treatment | Transpalatal Width Post-Treatment | Transpalatal Width Change | Percent Increase | *p*-Value |
|---|---|---|---|---|
| 32.9 ± 3.1 | 35.2 ± 3.2 | 2.3 ± 1.4 | 7.0% | $p < 0.00001$ |

Ten (10) out of 75 patients with pre- and post-treatment CBCT airway volumes (13%) showed a decrease in airway volume, while the rest of patients showed an increase or no change in airway volume (87%) (Table 5).

**Table 5.** Airway Volume (CBCT mm$^3$).

| Airway Volume Pre-Treatment | Airway Volume Post-Treatment | Airway Change | Percent Increase | *p*-Value |
|---|---|---|---|---|
| 22,263 ± 8651 | 25,567 ± 9550 | 3304 ± 4886 | 15% | *p* < 0.00001 |

*3.2. Sleep Study Results*

Table 6 (below) and Table 7 (below) shows the pre- and post-treatment changes in AHI produced by DNA treatment and numbers of patients by OSA classification respectively.

**Table 6.** AHI Changes.

| AHI Pre-Treatment | AHI Post-Treatment | AHI Change | Percent Decrease | *p*-Value |
|---|---|---|---|---|
| 22.5 ± 19.5 | 12.3 ± 11.0 | −10.3 ± 15.1 | 46% | *p* < 0.00001 |

**Table 7.** Number of Patients by OSA Classification.

| OSA Classification | Pre-Treatment | Post-Treatment No OSA | Post-Treatment Mild | Post-Treatment Moderate | Post-Treatment Severe |
|---|---|---|---|---|---|
| Mild | 42 | 18 | 19 | 5 | 0 |
| Moderate | 23 | 5 | 13 | 3 | 2 |
| Severe | 22 | 1 | 5 | 13 | 3 |
| Total | 87 | 24 | 37 | 21 | 5 |

Eighty percent of the patients showed some improvement in their AHI score, while 9% worsened and 11% showed no change (i.e., a <4.6% increase is within variability for events/hour in AHI sleep testing) [26]. Fifty-five percent of these patients improved by at least 1 classification, for example, from severe to moderate or moderate to mild. Finally, 24 (28%) of these patients had their OSA symptoms completely resolve (AHI < 5).

Prior to treatment, 22 patients had severe OSA, which improved (post-treatment) or was improving (midtreatment) by 53%. After treatment, 3 out of these 22 patients still had severe OSA (one worsened), 13 had moderate OSA, 5 had mild OSA, and 1 had no OSA.

Of the 23 patients that began the study with moderate OSA, the average percent improvement was 48%. Two patients' AHI score worsened by 1 category (10.4 and 8.1), while another slightly increased by 2.1 within the margin of AHI test errors. The balance of the patients improved by 60%. After treatment or midtreatment, 3 patients remained moderate, 13 had mild symptoms, and 5 had resolved their OSA symptoms.

Forty-two (42) patients had mild AHI symptoms pretreatment. Of these, none developed severe symptoms, five had moderate symptoms (1 had a 0.5 increase), 19 retained mild symptoms, and 18 had no OSA. The average percent improvement was 23%. Excluding those that worsened, the rest of the patients had an average improvement of 37%. Additionally, data were evaluated to see if correlations existed between AHI and patient characteristics and compliance. We found several non-significant correlations between changes in AHI and changes in transpalatal width (0.112), and the change in AHI was positively (albeit not significantly) correlated with changes in airway volume (−0.096). Additionally, changes in AHI were also negatively (not significantly) correlated with initial patient BMI (−0.210), and the change in AHI was negatively correlated with both age (−0.291) and hours worn (−0.159) (all not significant).

### 3.3. Adjunctive Modalities

In the treatment process, several adjunctive modalities were used. Adjunctive therapies were excluded if the researchers believed that the therapy would have a long-term effect on AHI. Excluded therapies included sinus surgery, tonsillectomy, laser therapy, and chiropractic care. The three adjunctive therapies included in this patient set were (i) myofunctional therapy to train the tongue to lay flat in the expanding palate; (ii) CPAP to train tongue position and be used during the initial treatment stage, primarily if the patient was already using CPAP or if the dentist was concerned about severe sleep apnea, clinical symptoms, or patient safety; and (iii) tongue tie releases or frenectomies.

The expansion device was used with no other modalities in 51 (59%) patients, who showed an average improvement of 7.1 (37%) in AHI scoring. The appliance was also used in 10 patients with myofunctional therapy, who showed an improvement of 18.2 (75%). CPAP was only used in conjunction with the appliance in 12 patients, who showed an average improvement in AHI scores of 16.9 (43%). Myotherapy with releases were used in conjunction with the appliance in another 9 patients, resulting in an AHI improvement of 7.5 (38.7%), almost identical to the appliance alone. Tongue tie releases and frenulectomies showed less improvement in AHI scores (6.4 and 6.5, respectively). The best results were achieved when the appliance was used with myofunctional therapy, where 80% of these patients resolved their AHI symptoms. The appliance showed only minor improvements when used with CPAP. These results were not unanticipated because both CPAP and myofunctional therapy train the patient to breathe properly with the appliance. All sleep tests were performed without any appliance in the mouth, so those results may reflect both persistent improvements that influenced the expansion of the jaw through appliance therapy and enhanced training induced by CPAP and myofunctional therapy. However, we recognize that tongue tie releases, frenulectomies, and myotherapies can have therapeutic effects that are captured in sleep studies, and in such cases we cannot state absolutely that OSA improvements are confined to jaw expansion.

### 3.4. Safety Results

Participating dentists reported six patients (6.9%) with an open bite, five caused by the appliance (5.8%) and one caused by tongue thrusting, a forward positioning of the tongue during rest, also known as orofacial myofunctional disorder (OMD), which can provoke open bite. Two of the open bites caused by the appliance were treated successfully with orthodontia, including aligners. One patient had such a small open bite that no treatment was necessary. One patient (1%) had mild gum recession caused by either the medication or the device losing canine guidance, which was adjusted at the next visit so ultimately no treatment was required. One patient reported pain, which resolved when the appliance was adjusted. One patient suffered secondary traumatic occlusion, resulting in loose teeth, which the dentist believed would completely resolve with post-treatment aligner therapy. One patient with poor hygiene before and during treatment reported tissue swelling. None of these complications had affected the patient's satisfaction with the outcome or caused serious health or dental issues.

## 4. Discussion

After treatment with the DNA, on average, the improvements in both transpalatal width and airway volume were remarkable and highly statistically significant with a *p*-value < 0.00001. After treatment with the DNA, transpalatal width increased by 7% (Table 4) and airway volume increased by 15% (Table 5). Additionally AHI scores improved by 46% ($p < 0.00001$) (Table 6). 91% of the patients improved or stayed the same, with 63% improved by 1 classification, and 28% of patients had their OSA completely resolved (Figure 3 and Table 7). In total, 51% patients either improved at least 45% or had their AHI resolved, exceeding a target used by sleep physicians for successful treatment.

Because airway resistance can be related to the 4th power of the radius of the cross-sectional area, the 15% increase in airway volume (assuming no change in the length of

the airway) would be expected to reduce airway resistance by 75%. These improvements dramatically reduce airway resistance and lower the frequency at which AHI/snoring occurs, while also reducing risks from OSA-associated comorbid conditions. Because these effects appear to be long-lasting, this could represent a permanent anatomic/functional solution for OSA.

These findings are comparable to the literature for mandibular advancement devices (MAD). For example, one study by Duran-Cantolla et al. reported, "The findings of this study indicated that 10.3% of the patients with MAD had their AHI increased by 50%, in comparison to the 31.6% in the case of the PD (placebo device)" [27]. Similarly, another study reported a $52.8 \pm 39.1\%$ improvement in AHI scores with MAD treatment [28]. When compared to prior studies with rapid maxillary expansion (RME) treatment, the DNA also performed comparatively well. One study found a reduction in AHI scores from $19 \pm 4$ to $7 \pm 4$ [29], while another RME treatment study reported an average reduction of 65.3% in the AHI [30].

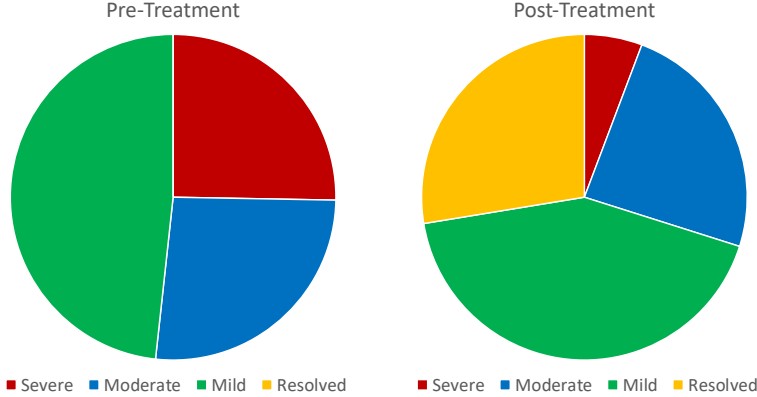

**Figure 3.** OSA severity classifications of patients before and after treatment, reflecting the values shown in Table 7. Resolved, or no OSA, is identified as AHI $\leq 5$; mild is AHI $\geq 5$; moderate is AHI $\geq 15$; and severe is AHI $\geq 30$.

We do, however, acknowledge several limitations within this study. While real world data are not as controlled as randomized clinical trials, they can still adequately measure the safety and efficacy endpoints obtained in an average dental practice. An evaluation of accrual data showed that all dentists were trained under the same protocol and given the same informed consent and data capture form. However, the protocol was subject to the dentist using his/her professional judgement to determine the best care for each patient and the customization of the device. Bias was minimized because each dentist submitted data to the research database without knowing how the data would be used. Potential bias was also minimized by the geographic diversity of site locations from all over the United States and a small portion of Canada (38 sites/doctors). Another factor minimizing any bias was that the primary AHI score was interpreted by sleep specialists who were independent from the clinical trial. Sleep studies were read by independent sleep specialists who might have used either a 3% or 4% scoring for hypopneas. This difference is mitigated by the pre- and post-treatment measurements being read by the same specialist using the same AASM criteria for the paired *t*-tests [31]. Recording procedures into the same database also forced consistency, as did the standardization of the measurement protocol for AHI and other measurements. For the question of relevance, the data set had a similar balance between male and female subjects to the predicate data set, an age range from 19–84, and representation of all ethnicities. It should also be noted that the patients had to be covered by insurance or be able to pay for the treatment, so patients living in poverty were not likely to be included in the database. Since the 38 sites were distributed primarily across the United States, the data could be generalized to the US population. Additionally, an analysis of 'poolability' showed that the data from each site could be statistically merged.

## 5. Conclusions

The analysis of this real-world data showed a 46% decrease in AHI ($p < 0.00001$), which is comparable to the findings in all previous studies, which show a 52% average decrease in AHI. Excluding two patient outliers from the dataset showed that the average AHI improvement was actually 54%. By treating the root cause of the problem and expanding the maxilla instead of using continuous positive air pressure or mandibular advancement, 28% of these patients had their OSA symptoms completely resolved. Additionally, expansion data confirms that the device does, in fact, increase airway volume and transpalatal width ($p < 0.00001$). As a non-surgical maxillary expansion device, the DNA presents a valuable alternative to mandibular advancement devices for the treatment of OSA. It is highly likely that these changes will be sustained over time. In fact, some patients that have provided data for more than five years out from these studies demonstrate that long-term resolution is possible. Further studies must be conducted to confirm that these changes are permanent in larger clinical trials. If these changes prove to be long-term, then DNA treatment might be more advantageous in certain patients than both CPAP and MAD as it eliminates the need for long-term patient compliance, which is the major cause of treatment failure for both therapies.

**Author Contributions:** Conceptualization, C.C., C.K. and J.G., methodology, C.C.; formal analysis, C.C.; investigation, N.D., C.C., C.K., J.G. and J.S.A.; resources, N.D., C.C., C.K., J.G. and J.S.A.; data curation, C.C.; writing—original draft preparation, N.D., C.C., C.K., J.G., O.C. and J.S.A.; writing—review and editing, N.D., C.C., C.K., J.G., O.C. and J.S.A.; visualization, C.C.; project administration, C.C. and J.S.A.; All authors have read and agreed to the published version of the manuscript.

**Funding:** This research received no external funding.

**Institutional Review Board Statement:** After IRB exemption by the Program for Protection of Human Subjects at the Icahn School of Medicine at Mount Sinai (STUDY-21-01561), we conducted a retrospective review of the Vivos Airway Intelligence Service (AIS) database, a prospectively collected clinical database. Patient data were anonymized for use by the research group.

**Informed Consent Statement:** All dentists were provided with informed consent forms for all participating patients.

**Conflicts of Interest:** Dr. Clete Kushida is a member of the medical advisory board at Vivos Therapeutics, Dr. Jonathan Greenberg serves as a member of the medical advisory board at Vivos, Dr. Colette Cozean serves as a regulatory consultant for Vivos, and Dr. Jonathan Steven Alexander is funded to accomplish research on Vivos products. None of the authors had any role in the design of or collection of the study database (Vivos Airway Intelligence Service (AIS) database, a prospectiv51kkely collected clinical database) used for this research.

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
