# Peer review of "Retrospective Analysis of Real-World Data for the Treatment of Obstructive Sleep Apnea with Slow Maxillary Expansion Using a Unique Expansion Dental Appliance (DNA)"

_pathophysiology, doi:10.3390/pathophysiology30020017_

Round 1

Reviewer 1 Report

Introduction

Line 52: What is meant by “oral appliances represent the most accessible option”. What does accessibility mean? Are you referring to the Phillips recall that has made it difficult to get CPAP machines, i.e. less accessible? If so, you could explicitly state this.

Line 70: The study by Vinha is in adults. It is a bit confusing because it seems like you are describing effect of RME in children and setting up the problem statement to say that we haven’t formally investigated effect RME on sleep disordered breathing in adults. I think a more accurate framing is that RME is effective in children, there is potential in adults based on results of surgically assisted RME. No one has investigated the RME (without surgical assistance) in adults.  

Methods

Line 147: Please indicate which AASM criteria were used to score the sleep studies.

Line 151: Can you provide a brief summary of how transpalatal width of the jaw is measured, for reproducibility. A reference would also be useful.

Line 163: You state CBCT would measure  airway volume at narrowest point of the patient’s mouth. First, do you mean narrowest part of the airway, i.e. velopharynx/oropharynx? Second, if you are measuring dimension at a point, then this would be an airway area, not a volume. You need to consider some length along the airway in order to compute a volume.   

Line 169: List of endpoints is numbered 1, 6, 7, 8

Line 175: If you know the sample size was sufficient then you presumably performed a power analysis. If so, can you report the results of the power analysis (i.e. power analysis showed that a sample size of X was able to detect changes in AHI, transpalatal width, and a airway volume of X, Y, and Z, respectively, at power level of X and alpha level of 0.05.  

LRegarding adjunct therapies: You excluded patients on certain adjunctive therapies (e.g. tonsillectomy), but not others (CPAP, myotherapy, etc.). It is not clear to me how this decision was made. Also, it is not explicitly stated that patients on adjunctive therapies were included in the analysis (Line 130 says patients were offered adjunctive therapy, but its not clear how they were treated in the analysis). Reading the results made that clear, but having said all that, I am not sure it is appropriate to include patients on any adjunctive therapy (see comment in results about this).

Results:

Table 1 needs units for age and BMI

Line 217: What about baseline transpalatal width and airway volume? Was change in transpalatal width or airway volume associated with treatment response?  Also, BMI is typically body-mass index, not basal metabolic index. Is this an error or did you actually calculate basal metabolism?

Line 244: what is tongue thrusting?

Regarding adjunctive therapy: I think it is misleading to include patients with adjunctive therapies in the sample because then you are no longer isolating for the effect of RME, you are investigating the effect of RME with and without adjunctive therapy on OSA. Given that you will still have 51 patients, your should will remain significant, and I think making this change will strengthen the study. I would still report on the patients that had adjunctive therapies separately since it is very interesting (especially the myotherapy group), but it should not be your primary finding. Even though you say that all sleep tests were performed without appliances in the mouth such that improvements are isolated to be from expansion, this is not necessarily true because the tongue tie release, frenulectomies and myotherapies will have therapeutic effect that will be captured in the sleep study (i.e. you can’t undo the effects for the treatment sleep studies). Therefore, in these cases in particular, you cannot say that OSA improvements are isolated to jaw expansion. CPAP is less problematic, but there is evidence that OSA is less severe in patients who are on CPAP that get tested without their CPAP. It takes 3 or more days untreated to return to “baseline” levels of OSA severity.

Discussion:

Should be updated based on the results from patients without adjunctive therapies.

Author Response

Reviewer 1. Comments and Suggestions for Authors

Introduction

Line 52: What is meant by “oral appliances represent the most accessible option”. What does accessibility mean? Are you referring to the Phillips recall that has made it difficult to get CPAP machines, i.e. less accessible? If so, you could explicitly state this.

Response: We have substituted “popular” for “accessible”.

Line 70: The study by Vinha is in adults. It is a bit confusing because it seems like you are describing effect of RME in children and setting up the problem statement to say that we haven’t formally investigated effect RME on sleep disordered breathing in adults. I think a more accurate framing is that RME is effective in children, there is potential in adults based on results of surgically assisted RME. No one has investigated the RME (without surgical assistance) in adults.  

Response: We agree with the reviewer and have modified the statements.

Methods

Line 147: Please indicate which AASM criteria were used to score the sleep studies.

Response: We now state that:

Sleep studies were also read by an independent sleep specialist, who might have used either a 3% or 4% scoring for hypopneas. This difference is mitigated because the pre- and post-treatment measurements are read by the same specialist with the same AASM criteria for the paired t-tests [31].

Line 151: Can you provide a brief summary of how transpalatal width of the jaw is measured, for reproducibility. A reference would also be useful.

Response: We have defined our measurement technique in the article.

Line 163: You state CBCT would measure  airway volume at narrowest point of the patient’s mouth. First, do you mean narrowest part of the airway, i.e. velopharynx/oropharynx? Second, if you are measuring dimension at a point, then this would be an airway area, not a volume. You need to consider some length along the airway in order to compute a volume.   

Response: We have defined our measurement technique in the article.

Line 169: List of endpoints is numbered 1, 6, 7, 8

Response: This has been corrected.

Line 175: If you know the sample size was sufficient then you presumably performed a power analysis. If so, can you report the results of the power analysis (i.e. power analysis showed that a sample size of X was able to detect changes in AHI, transpalatal width, and a airway volume of X, Y, and Z, respectively, at power level of X and alpha level of 0.05.  

Response: The sample size estimation has been added.

Regarding adjunct therapies: You excluded patients on certain adjunctive therapies (e.g. tonsillectomy), but not others (CPAP, myotherapy, etc.). It is not clear to me how this decision was made. Also, it is not explicitly stated that patients on adjunctive therapies were included in the analysis (Line 130 says patients were offered adjunctive therapy, but its not clear how they were treated in the analysis). Reading the results made that clear, but having said all that, I am not sure it is appropriate to include patients on any adjunctive therapy (see comment in results about this).

Response: A clarifying statement has been added.

Results:

Table 1 needs units for age and BMI

Response: Done

Line 217: What about baseline transpalatal width and airway volume? Was change in transpalatal width or airway volume associated with treatment response?  

Also, BMI is typically body-mass index, not basal metabolic index. Is this an error or did you actually calculate basal metabolism?

Response: We calculated body mass index; this has been corrected.

Line 244: what is tongue thrusting?

Response: This has been clarified.

Regarding adjunctive therapy: I think it is misleading to include patients with adjunctive therapies in the sample because then you are no longer isolating for the effect of RME, you are investigating the effect of RME with and without adjunctive therapy on OSA. Given that you will still have 51 patients, your should will remain significant, and I think making this change will strengthen the study. I would still report on the patients that had adjunctive therapies separately since it is very interesting (especially the myotherapy group), but it should not be your primary finding. Even though you say that all sleep tests were performed without appliances in the mouth such that improvements are isolated to be from expansion, this is not necessarily true because the tongue tie release, frenulectomies and myotherapies will have therapeutic effect that will be captured in the sleep study (i.e. you can’t undo the effects for the treatment sleep studies). Therefore, in these cases in particular, you cannot say that OSA improvements are isolated to jaw expansion. CPAP is less problematic, but there is evidence that OSA is less severe in patients who are on CPAP that get tested without their CPAP. It takes 3 or more days untreated to return to “baseline” levels of OSA severity.

Response: We concur that you have a valid point in changing the analysis.  However, this study was conducted with collaboration from the FDA, who insisted that since the Vivos Method (the protocol taught to the dentists), allows for certain adjunct therapies, they should be the standard of care. Further, these modalities (CPAP, surgical releases and myofunctional therapy) in and of themselves don’t significantly affect the airway size and AHI.  Therefore it was requested that the data be presented for all modalities together and then provide a stratified analysis of DNA treatment with each of the modalities.  As you state, the data for DNA alone is also presented. 

Discussion: Should be updated based on the results from patients without adjunctive therapies.

Please see our previous response.

Reviewer 2 Report

Dear Author (s)

1. There are several grammatical errors. 

2. Please match tables together. For example, Table 1 with 2, 3.

3. The samples are low.

4. Yu used one variable for each table. It is not interesting.

5. Figure 2 is additional figure. It is similar to Table 7.

6. The analysis is sample.

7. Where are ethical code and grant number?

8. Some section of results need tables and graphs.

9. Please use recent citations more in text, especially introduction.

Author Response

We appreciate the helpful comments.

1. There are several grammatical errors. 

Response: These have been corrected.

  1. Please match tables together. For example, Table 1 with 2, 3.

Response: Thanks for your comment; we could not readily combine Tables 1-3 since they were displaying different data.

  1. The samples are low.

Response: We have added the sample size estimation in Section 2.6.

  1. Yu used one variable for each table. It is not interesting.

Response: We believe that combining the variables in a single table will detract from the clarity of the presented data.

  1. Figure 2 is additional figure. It is similar to Table 7.

Response: We believe that although the data in Figure 2 and Table 7 are similar, Figure 2 represents the proportion of patients pre- and posttreatment in the different OSA severity classes.

  1. The analysis is sample.

Response: Although the statistical analysis is simple, the outcomes were highly significant.

  1. Where are ethical code and grant number?

Response: The IRB exemption number is in Section 2.1; the study was funded by Vivos Therapeutics.

  1. Some section of results need tables and graphs.

Response: We believe that we have presented the principal results in tabular and graphical forms; the rest of the data presented are secondary and we feel don’t warrant presentation in tables and figures.

9. Please use recent citations more in text, especially introduction.

Response: We believe that we have now included all relevant citations.

Round 2

Reviewer 1 Report

Line 52. Not to be pedantic, but I still do not understand when you say oral appliance are the most popular for patients. Especially when you are saying they are more popular than CPAP. In the sentence before you say CPAP is the most common therapy.  How can CPAP be most common, but oral appliance be more popular? Things that I know to be true are that oral appliances are the most commonly used non-PAP alternative and that adherence to oral appliance exceeds that of CPAP. I am not sure of anyone that have defined or quantified popularity of oral appliance vs. CPAP. 

Line 79. I cannot interpret the highlighted section addressing my comment. I am not sure if there was a problem in converting the document into PDF but I cannot understand the sentence starting on line 79.

Line 241 you state correlations done with AHI. Where are the results?

I am still not clear on the volume measurement of the airway. I see you made a clarifying statement that the area was measured at the narrowest point in the velopharynx. I do not see any area measurements in the results (i.e., Table 5 says volume and has units of mm3). How does this area measurement relate to the volume measurement presented in Table 5. Also, you say narrowest point of that patient's mouth, but then in parentheses say velopharynx. I would not describe the velopharynx as being part of the mouth. Can you clarify using anatomical landmarks to denote the boundaries within which the narrowest point was identified. 

Author Response

Line 52. Not to be pedantic, but I still do not understand when you say oral appliance are the most popular for patients. Especially when you are saying they are more popular than CPAP. In the sentence before you say CPAP is the most common therapy.  How can CPAP be most common, but oral appliance be more popular? Things that I know to be true are that oral appliances are the most commonly used non-PAP alternative and that adherence to oral appliance exceeds that of CPAP. I am not sure of anyone that have defined or quantified popularity of oral appliance vs. CPAP. 

Response. We appreciate this and now state:

Oral appliances remain the most commonly used alternative to positive airway pressure approaches and adherence to oral appliance exceeds that of CPAP reflecting the mechanical complexity, discomfort and high maintenance associated with CPAP. 

Line 79. I cannot interpret the highlighted section addressing my comment. I am not sure if there was a problem in converting the document into PDF but I cannot understand the sentence starting on line 79.

Response. This was an error and now reads as:

One study by Vinha et al., 2018 evaluating surgically assisted rapid maxillary expansion on obstructive sleep apnea and daytime sleepiness found that this approach significantly reduced the respiratory disturbance index by an average of 54.6 % (p = 0.0013). Because adult oral anatomy is assumed to be ‘fixed,’ most expansion appliance studies in adults use surgically assisted RME which although more rapid, has the potential for pain requiring pain management and requires surgery [23,24].

Line 241 you state correlations done with AHI. Where are the results?

Response. We now state:

We found several non-significant correlations between changes in AHI and changes in transpalatal width (0.112), the change in AHI was positively (albeitly not significantly) correlated with changes in airway volume (-0.096). Additionally changes in AHI were also negatively (not significantly) correlated with initial patient BMI (-0.210) and the change in AHI was negatively correlated with age (-0.291) and negatively correlated with hours worn (-0.159)(all not significant).

I am still not clear on the volume measurement of the airway. I see you made a clarifying statement that the area was measured at the narrowest point in the velopharynx. I do not see any area measurements in the results (i.e., Table 5 says volume and has units of mm3). How does this area measurement relate to the volume measurement presented in Table 5. Also, you say narrowest point of that patient's mouth, but then in parentheses say velopharynx. I would not describe the velopharynx as being part of the mouth. Can you clarify using anatomical landmarks to denote the boundaries within which the narrowest point was identified. 

Response. We appreciate this as well and have added a figure and description which explains the anatomical regions being evaluated.

For CBCT measurements, airway cross-sectional area was measured at the narrowest point of the patient’s upper airway (see Figure 2). Airway volume is measured at a defined region of the pharynx (excluding nasal passages). A 3D voxel selection is made with in a specified boundary around the pharynx with inferior airway border parallel to the floor with anterior-inferior border of C3 and following along anterior nasopharynx. Length is from top of pharynx to inferior border of C3. 

Reviewer 2 Report

Dear

The manuscript is acceptable.

Author Response

We appreciate the support of the reviewer.